# Estimating the effectiveness of non-pharmaceutical interventions against COVID-19 transmission in the Netherlands

**Jantien A. Backer**[1]*, **Don Klinkenberg**[1], **Fuminari Miura**[1,2], **Jacco Wallinga**[1,3]

**1** Centre for Infectious Disease Control, National Institute of Public Health and the Environment: Rijksinstituut voor Volksgezondheid en Milieu, Bilthoven, the Netherlands, **2** Center for Marine Environmental Studies (CMES), Ehime University, Ehime, Japan, **3** Biomedical Data Sciences, Leiden University Medical Center, Leiden, the Netherlands

\* jantien.backer@rivm.nl

## Abstract

During the COVID-19 pandemic non-pharmaceutical interventions (NPIs) were taken to mitigate virus spread. Assessing their effectiveness is essential in policy support but often challenging, due to interactions between measures, the increase of immunity, variant emergence and seasonal effects. These factors make results difficult to interpret over a long period of time. Using a mechanistic approach, we estimate the overall effectiveness of sets of NPIs in reducing transmission over time. Our approach quantifies the effectiveness by comparing the observed effective reproduction number, which is the number of secondary infections caused by a typical infected person, to a counterfactual reproduction number if no NPIs were taken. The counterfactual reproduction number accounts for seasonal variations in transmissibility, for emergence of more transmissible variants, and for changes in immunity in the population. The immune fraction is reconstructed from age-specific data of longitudinal serological surveys and vaccination coverage, taking immunity loss due to waning into account. We estimate the effectiveness of NPIs in the Netherlands from the start of the pandemic in March 2020 until the emergence of the Omicron variant in November 2021. We find that the effectiveness of NPIs was high in March and April 2020 during the first pandemic wave and in January and February 2021, coinciding with the two periods with the most stringent measures. For both periods the effectiveness was estimated at approximately 50%, i.e., without any measures the reproduction number would have been twice as high as observed. The proposed approach synthesises available epidemiological data from different sources to reconstruct the population-
level immunity. With sufficient data, it can be applied not only to COVID-19 but also to other directly transmitted diseases, such as influenza. This method provides a near real-time assessment of the effectiveness of control measures when the required data are available.

**Data availability statement:** All data is publicly available, either as open data or as additional data sets on the github repository accompanying this study: github.com/rivm-syso/effectiveness_NPIs. This repository also contains the code used to produce the results and analyses presented in this manuscript.

**Funding:** This study was funded by the Ministry of Health, Welfare and Sport (VWS) in the Netherlands (https://www.rijksoverheid.nl/ministeries/ministerie-van-volksgezondheid-welzijn-en-sport) (JB, DK, FM and JW). FM and JW received funding from European Union's Horizon research and innovation programme - project ESCAPE (Grant agreement number 101095619, https://www.escapepandemics.com/). FM was supported by the Ministry of Education, Culture, Sports, Science and Technology, Japan (MEXT, https://www.mext.go.jp/en/) to a project on Joint Usage/Research Center – Leading Academia in Marine and Environmental Pollution Research (LaMer). FM acknowledges fundings from Japan Society for the Promotion of Science (JSPS KAKENHI, 20J00793, https://www.jsps.go.jp/english/e-grants/) and Japan Science and Technology Agency (JPMJPR23RA, https://www.jst.go.jp/kisoken/presto/en/). The funding agencies did not play any role in the study design, data collection and analysis, decision to publish, nor preparation of the manuscript.

**Competing interests:** The authors have declared that no competing interests exist.

## Author summary

During the COVID-19 pandemic, many interventions were taken aimed at reducing contacts between people, such as school closures and working-from-home recommendations. At the same time, the reproduction number - denoting the number of infections one infected person can cause - became a well-known term. We asked what the value of the reproduction number would have been in absence of interventions. This value depends on factors such as the number of persons that is infected or vaccinated over time, immunity waning, seasonal effects, and the emergence of more transmissible variants. By combining multiple data sources from the Netherlands, we accounted for these factors and calculated the reproduction number without any interventions. We found that during the two periods with the most stringent measures in the Netherlands the reproduction number without interventions would have been twice as high as the observed effective reproduction number with interventions, indicating an effectiveness of about 50% of the measures at that time. A similar approach to estimating the effectiveness of intervention measures can be applied to COVID-19 data of other countries or to other directly transmitted diseases.

## Introduction

In most countries worldwide a great variety of non-pharmaceutical interventions (NPIs) were taken during the COVID-19 pandemic, ranging from hand-washing recommendations to strict stay-at-home orders. Especially before the introduction of vaccination, these measures were paramount to curb virus transmission. Understanding how effective the different sets of NPIs were, will help to be better prepared for future pandemics.

Many studies have been published on the effectiveness of NPIs [1,2]. A natural outcome measure for assessing effectiveness is the effective reproduction number [3], which is the average number of secondary infections caused by one infected person. Existing studies use parametric regression analyses to estimate the effectiveness of individual measures by combining data on reproduction numbers and NPIs from different regions [4–9]. However, these methods assume that the effectiveness of each measure is time-invariant and independent of other measures. In practice, interactions between measures exist [10] and the effectiveness can vary over time, underlining that there is no constant effectiveness to a single measure.

Instead of attempting to isolate the effect of a specific measure, we focus on the overall effectiveness of sets of NPIs in different periods of the pandemic, using a mechanistic approach that takes the time-varying immunity, seasonality and variant emergence into account. To do so, we compare the effective reproduction number [11,12], estimated from incidence data, to a counterfactual reproduction number that would have been expected in the same (partially immune) population, at the same time during the pandemic, but without any interventions. Many studies take the relative reduction of the reproduction number as a measure for effectiveness

[3], although almost all studies focus on the first pandemic wave [13,14]. A measure of effectiveness should also reflect uncertainty in observations while maintaining a practically meaningful precision. Such characteristics allow for a fair and interpretable comparison of the effectiveness of all NPIs in different periods of the pandemic.

The challenge in obtaining precise and well-calibrated estimates of effectiveness of measures lies in estimating the counterfactual reproduction number. This reproduction number changes in two basic ways. Firstly, circumstances can change over time because of the emergence of more transmissible variants [15] or because of seasonal effects [9,16–18]. Secondly, an increase in the number of immune persons in the population would decrease the counterfactual reproduction number, and vice versa. The immune fraction increases because of infections and vaccinations, and it decreases due to waning of infection-induced (i.e., natural) and vaccine-induced immunity.

We aimed to quantify the overall effectiveness of sets of NPIs against transmission in the Netherlands from the start of the pandemic (March 2020) until the emergence of the Omicron variant (November 2021). We estimated the growth advantage of emerging variants from genomic surveillance data [19] and used the seasonal effect estimated for the Netherlands [18]. We reconstructed the immune fraction of the population over time using age-specific data of longitudinal serological surveys [20], test-positive cases [21,22] and vaccination coverage [23], assuming immunity waning rates from literature [24,25]. We obtained estimates of the effectiveness of NPIs against transmission during 28 periods with distinct sets of NPIs. To check the robustness of our conclusion, analyses were also performed with varying assumptions on immunity waning and seasonal effects. All data and code are published online [26].

## Results

We determined the effectiveness of NPIs over time $\psi(t)$ by comparing the observed effective reproduction number $R_{\text{eff}}(t)$ to a counterfactual reproduction number $R_c(t)$:

$$\psi(t) = 1 - \frac{R_{\text{eff}}(t)}{R_c(t)}.$$

The effective reproduction number $R_{\text{eff}}(t)$ was calculated from case data or hospitalisation data [27] in the Netherlands during the epidemic and published online [11,12]. The counterfactual reproduction number $R_c(t)$ reflects the transmission potential without any NPIs, defined as the basic reproduction number $R_0(t)$ reduced by the immune fraction of the population over time $\phi(t)$:

$$R_c(t) = (1 - \phi(t))\, R_0(t).$$

By using the population average of the immune fraction $\phi(t)$ we disregard any heterogeneities in, for instance, contact structure or transmission specific settings. The basic reproduction number $R_0(t)$ and the immune fraction $\phi(t)$ over time were estimated by combining different data sources (see the following sections and Methods).

### Change in basic reproduction number

The basic reproduction number $R_0(t)$ is defined as the average number of secondary infections caused by one infected person in a fully susceptible population without any interventions. We allow the basic reproduction number to vary over time depending on the circumstances at time $t$ [28,29], specifically on the time of year and on the circulating variants. We model the seasonal fluctuations $\sigma(t)$ and the relative transmissibility $\theta(t)$ of circulating variants as multiplicative effects on the basic reproduction number:

$$R_0(t) = R_0^{\text{initial}}\, \theta(t)\, \frac{\sigma(t)}{\sigma(t_{\text{ref}})},$$

where reference time $t_{\text{ref}}$ is chosen at 1 March 2020 when the population was fully susceptible, before NPIs and before the emergence of more transmissible variants ($\theta(t_{\text{ref}}) = 1$). At this time the basic reproduction number is equal to the initial reproduction number $R_0^{\text{initial}}$. For $R_0^{\text{initial}}$ we use the average effective reproduction number before 1 March 2020 which was observed to be 2 [11,12].

The basic reproduction number increased due to the emergence of more transmissible variants. The relative transmissibility of variants compared to the wildtype was estimated from genomic surveillance data as 34%, 23%, 39% and 106% for the Alpha, Beta, Gamma and Delta variants, respectively (Section 1 in S1 Text), which is similar to reported pooled estimates for these variants [15]. Although the Omicron variant that emerged at the end of 2021 is also more transmissible, its growth advantage is known to be attributable to immune escape rather than an increased basic reproduction number [30,31]. For this reason, the Omicron period is not included in the analysis.

Because of seasonal effects on the transmissibility of the virus, we adopted the estimate that the basic reproduction number in summer is 41% lower than in winter [18]. The combined effects of the initial reproduction number, variant emergence and seasonality on $R_0(t)$ are shown in Fig 1.

## Reconstructed immune fraction of the population

The counterfactual reproduction number also depends on the immune fraction of the population. This fraction increases over time due to (re-)infections and vaccinations, and decreases because of waning of immunity. As infection incidence and vaccination coverage differed by age group, we reconstructed the immune fraction in 8 age groups (0–9, 10–19, 20–29, 30–39, 40–49, 50–59, 60–69, 70+), and used their weighted average. The cumulative infection incidence is informed by a longitudinal serological survey in a representative sample of the Dutch population [20,32,33]. The cumulative infected fraction in each age group is tracked, counting all persons that have been infected at least once. In other words, reinfections will not increase the cumulative infected fraction. The fraction increases monotonically as fewer naive persons remain in the population (Fig 2). The vaccination coverage starts to increase as the vaccination programme was

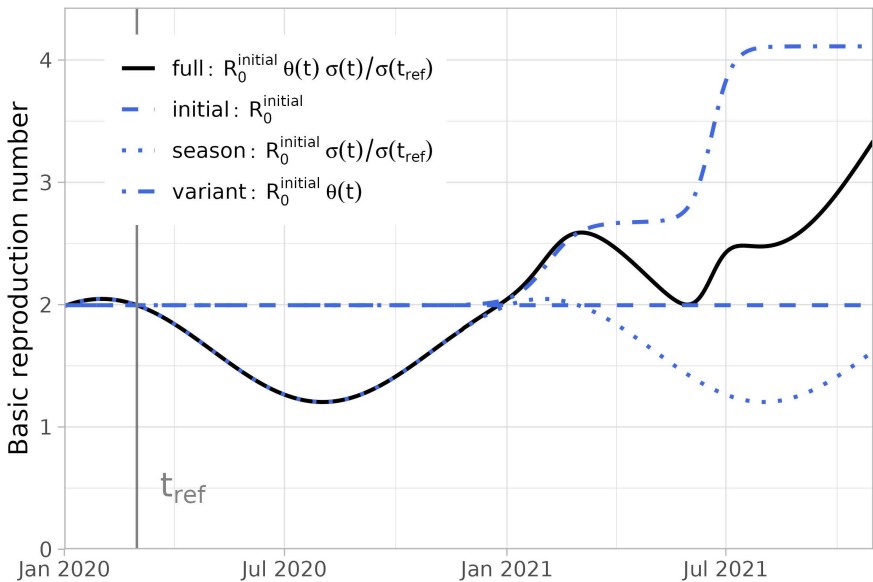

**Fig1. The basic reproduction number over time (1 January 2020–31 October 2021).** The basic reproduction number (full, black line) depends on the initial reproduction number observed at the start of the pandemic, seasonal effects and emergence of more transmissible variants (broken, blue lines). At reference date 1 March 2020 the basic reproduction number is equal to the initial observed reproduction number.

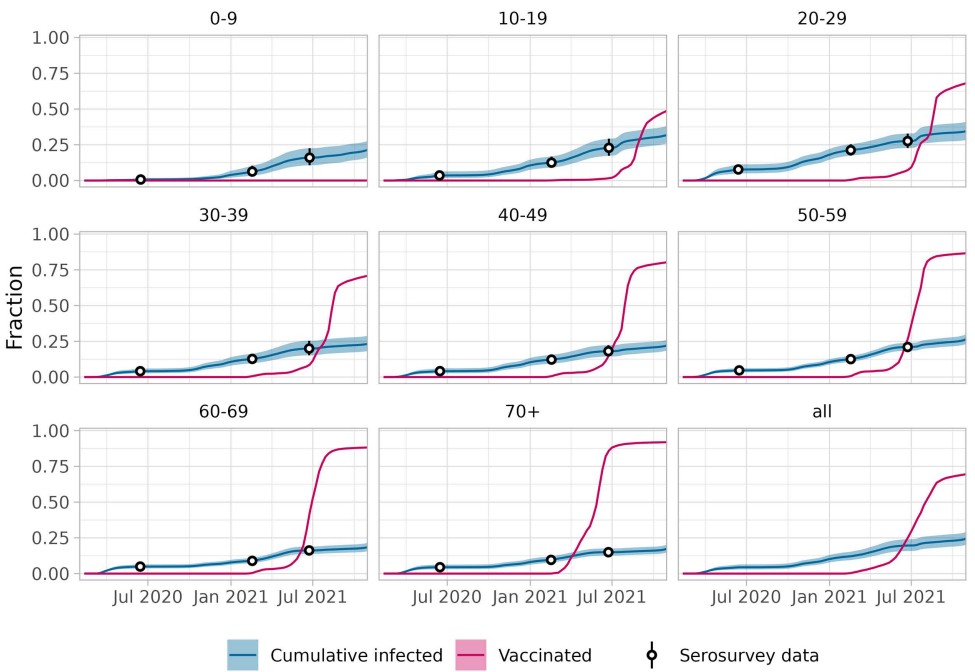

**Fig 2. Cumulative infected and vaccinated fractions by age group, and overall (1 February 2020–31 October 2021).** The cumulative infected fraction is reconstructed from serological surveys with a 95% confidence interval. The fraction is measured in survey rounds every few months; the fraction in between survey rounds is imputed using the daily cumulative number of test-positive cases per age group in between survey rounds (see Section 2.2 in S1 Text). The vaccinated fraction per day and age group was obtained by dividing the reported number of administered vaccinations per week and age group by the population size per age group and linear interpolation. These data were used as input to calculate the immune fraction per day and per age group.

rolled out in early 2021, starting in the older age groups as the vaccination priority was from old to young. The older age groups also achieved higher vaccination coverages (Fig 2).

The cumulative infected and vaccinated fractions are not necessarily immune, as immunity can wane over time. We reconstructed the immune fraction in each age group and in the full population over time using the data in Fig 2 and assuming immunity wanes exponentially over time, where natural immunity wanes faster than vaccine-induced immunity [24,25]. With these waning rates, the immune fraction is reconstructed from the infection and vaccination incidence on each day in each age group (see Methods and Section 2 in S1 Text). The population is separated in four subgroups: immune after infection, immune only by vaccination, those that lost their immunity (non-naive susceptibles), and naive susceptibles (Fig 3). The first two subgroups make up the immune fraction $\phi(t)$, needed to calculate the counterfactual reproduction number.

In the first year the population mainly consisted of naive individuals and individuals protected by infection, as natural immunity wanes slowly and infections occurred recently. In the second year individuals were protected by vaccination, but also the non-naive susceptible class arose, because natural immunity had waned (see for instance the unvaccinated 0–9 age group) and because vaccine-induced immunity wanes faster than natural immunity (consistent with the assumed waning rates).

Since the immune fraction depends on the waning rate, we evaluated three scenarios for waning rates of natural and vaccine-induced immunity that were taken from literature [24,25]. The reconstruction for the medium waning scenario is shown in Fig 3; the figures for slow and fast waning scenarios are given Fig C and D in S1 Text.

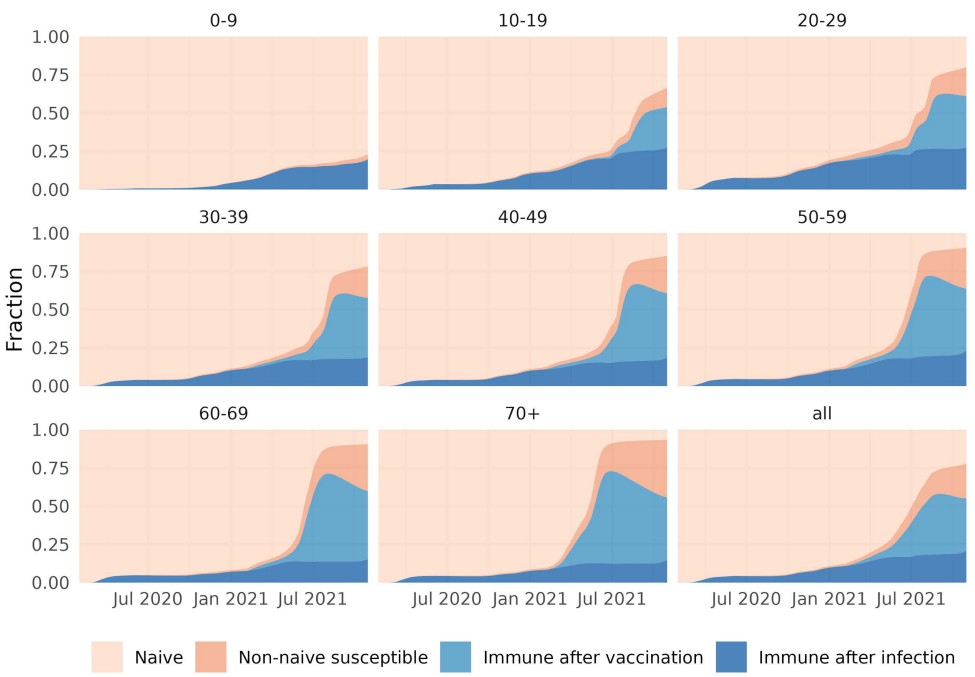

**Fig 3. Reconstructed immune fractions by age group (and overall) over time (1 February 2020–31 October 2021), assuming medium waning rates for natural and vaccine-induced immunity and a medium vaccine effectiveness.** The population is stratified in a naive class that has never been infected nor vaccinated, a class immune after infection, a class immune after only vaccination and a non-naive susceptible class that lost its natural or vaccine-induced immunity. The equations for the reconstruction of immune fractions are given in Section 2 in S1 Text.

To validate the reconstruction of the immune fraction, we used the naive and non-naive susceptible classes to calculate reinfections (in previously infected persons) and breakthrough infections (in vaccinated persons) as fractions of new infections and compared these to data. Overall, both fractions increase as the non-naive susceptible class increases compared to the naive class (Fig 4). The fractions are largest for the fast waning scenario. The fraction of breakthrough infections increases faster than the fraction of reinfections, as vaccine-induced immunity wanes faster than natural immunity. The reconstructed fractions of reinfections and breakthrough infections show a similar time course as the data. Especially the reconstructed fraction of reinfections in the medium waning scenario matches the data, and at the end of the study period the reconstructed fraction of breakthrough infections reaches values that were observed in the data. The reconstructed fractions of reinfections and breakthrough infections per age group are provided in Fig E and F in S1 Text.

**Effectiveness of non-pharmaceutical interventions against transmission**

With the reconstructed immune fraction in the population and the basic reproduction number, the counterfactual reproduction number was calculated. Comparing this with the observed effective reproduction number gives the effectiveness of NPIs (Fig 5). At the start of the pandemic (March 2020) without any control measures the effectiveness was 0% by definition, as the observed initial reproduction number was equal to the basic reproduction number.

During the first pandemic wave (March/April 2020) the effectiveness is estimated at around 50%, suggesting that without measures the observed reproduction number would have been twice as high. Towards the summer of 2020, the effectiveness decreased in steps, coinciding with the relaxation of measures [34]. It drops to around 0%, when most leisure activities were allowed with restrictions, such as a maximum number of guests in restaurants.

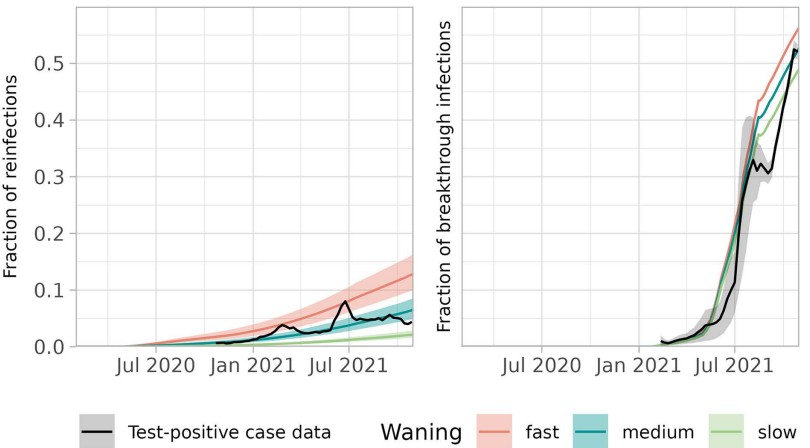

**Fig 4. Fraction of new infections that are reinfections (left) or breakthrough infections (right) for the full population, assuming slow, medium or fast waning (1 February 2020–31 October 2021).** Observations (black) are plotted for weeks where at least 50% of the test-positive cases has a known vaccination or infection status. The observed fraction of breakthrough infections (right panel) depends on whether infections in partly vaccinated persons are counted for 0% (lower limit of grey area), 50% (mean) or 100% (upper limit of grey area) as breakthrough infection. The observed fraction of reinfections (left panel) lacks this uncertainty.

In the first months of 2021 a long lockdown was imposed with more stringent measures than in the first pandemic wave [35]. The effectiveness of this set of measures is again estimated at around 50%. In this period the larger immune fraction due to infections and the first vaccinations was counteracted by the emergence of the more transmissible Alpha variant. As a result, the counterfactual reproduction number was higher than the reproduction number at the start of the pandemic.

In the beginning of July 2021, control measures were lifted on a large scale, including the reopening of night clubs [35]. As this coincided with the emergence of the Delta variant, a large wave of infections followed, and the effectiveness dropped below 0%, suggesting an adverse effect of the NPIs. This is caused by the observed effective reproduction number being higher than the counterfactual reproduction number, which suggests that more contacts were made than in the reference situation of March 2020. Possibly, the opening of night clubs and other leisure activities triggered a peak in human interactions. In the fall of 2021, a high vaccination coverage was achieved in all eligible age groups. Control measures were relatively light, with an estimated effectiveness of around 20%.

The different waning rate scenarios lead to similar estimates of effectiveness in the first year of the pandemic (Fig 5). Without vaccination and relatively few infections, the non-naive susceptible class is negligible for all waning rates (Fig 3). In the second year, the difference in the estimated effectiveness by waning scenario becomes larger mainly due to the large-scale vaccination roll-out (Fig 2).

The timing of changes in effectiveness is expected to correspond with changes in the Oxford stringency index [36] (shown as colour band in Fig 5). A positive change in the stringency index indicates that one or more measures were added or strenghtened, yielding an expected increase in effectiveness. Vice versa, a negative change indicates that one or more measures were lifted or alleviated, resulting in an expected decrease in effectiveness. Indeed, 22 out of 27 change points shown in Fig 5B are consistent with this expectation. In 2 instances the stringency increases and the effectiveness decreases, and in 3 instances the opposite is found.

The uncertainty in the estimated effectiveness varies over time. At the start of pandemic, the uncertainty of the estimate is large because the observed effective reproduction number was based on reported hospitalisations during this period. It increases even further because of the low numbers of hospitalisations after the first wave, yielding a uncertainty that spans a -50% to 100% range. From June 2020 onwards, the reproduction number was based on reported test-positive

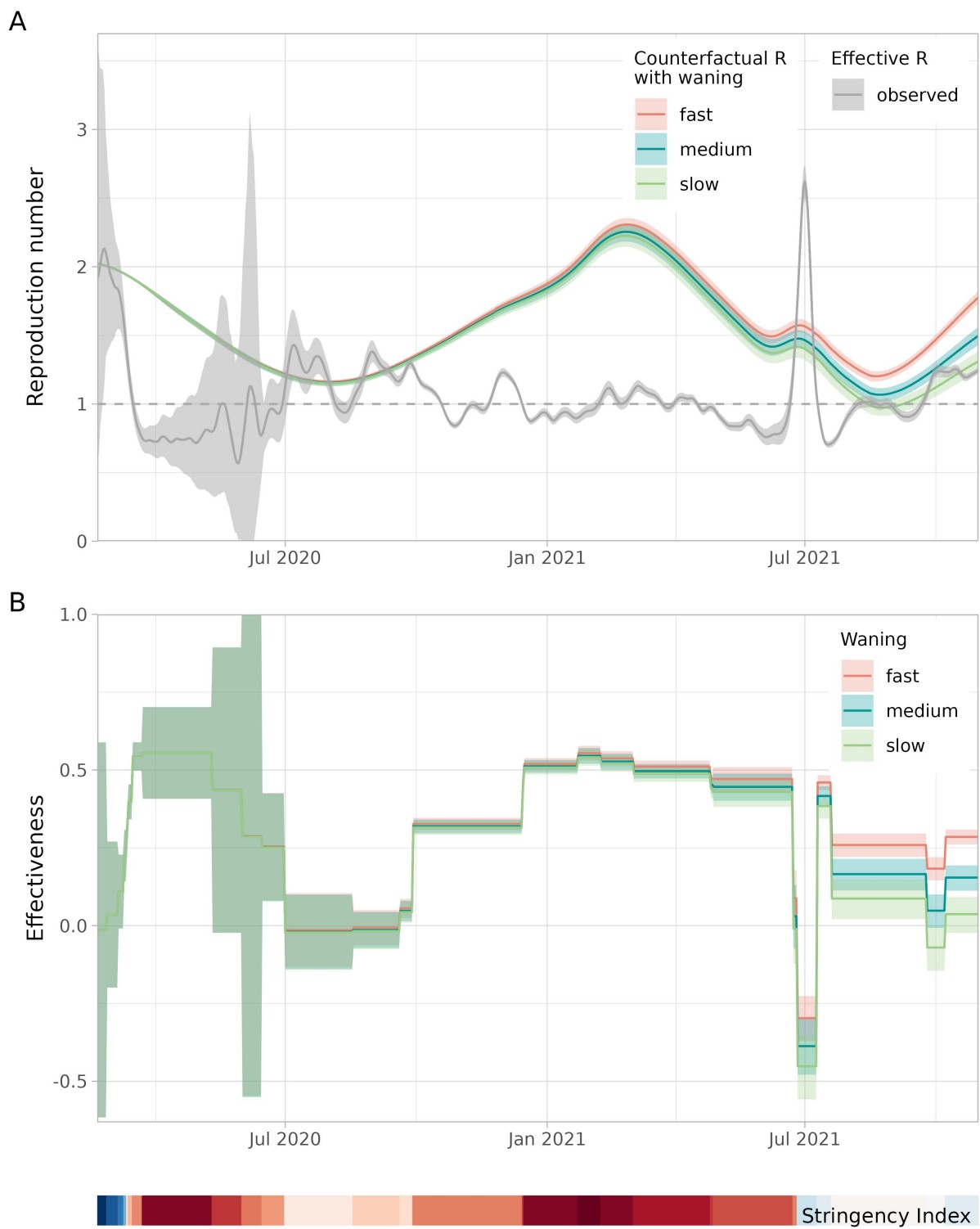

**Fig 5. Reproduction number and effectiveness of NPIs over time (20 February 2020–31 October 2021). A.** Counterfactual reproduction numbers assuming slow, medium or fast waning and the 7-day moving average of the effective reproduction number, based on hospitalisations (up to 13 June 2020) or test-positive cases (from 13 June 2020 onwards). **B.** The effectiveness of NPIs assuming slow, medium or fast waning, averaged over intervention periods with different sets of control measures (with constant stringency). As a timeline reference the Oxford Stringency Index [36] is depicted as a colour band from low stringency (darkest blue, index of 0) via intermediate (white, index of 41) to high (darkest red, index of 82).

cases, leading to more precise estimates. At the end of the study period the uncertainty of the estimates stems from uncertainty in the immune fraction and waning rates.

## Sensitivity analyses

In the main analysis, we assumed that the reproduction number is reduced by 41% in summer compared to winter [18]. When varying the amplitude of the seasonal changes in reproduction numbers, the results during summer are greatly affected but not those in winter (Fig 6). This is because the reference point lies on 1 March 2020, which anchors the peak of reproduction numbers in winter and only changes the decrease in summer. Without seasonality, the counterfactual reproduction number is much higher in summer which increases the estimated effectiveness, but even then the effectiveness remains lower in periods with lower stringency.

## Discussion

We estimated the overall effectiveness of NPIs against COVID-19 transmission over time in the Netherlands, by combining a range of data sources. The results reveal that NPIs reduced the reproduction number by about 50% during periods with the most stringent measures, with wide uncertainty bounds in April 2020 and narrow uncertainty bounds in March 2021. This estimate was consistent under different assumptions on seasonal effects or waning rates. The proposed

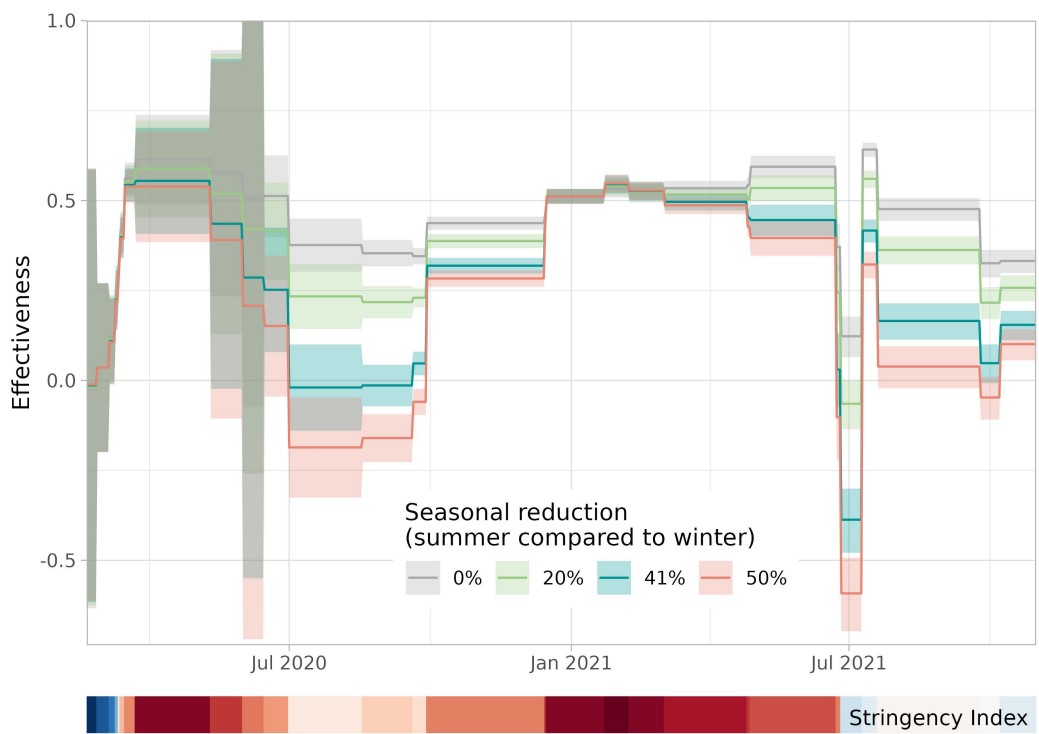

**Fig 6. Effectiveness of NPIs over time (20 February 2020–31 October 2021) for different assumptions of seasonality.** The reduction of the reproduction number in summer compared to winter in the main analysis is 41% [18]. The sensitivity analysis is carried out for values of 20% [16] and 50%. For completeness also the model without any seasonal effect is shown (reduction of 0%). The effectiveness of NPIs (assuming medium waning) is averaged over intervention periods with different sets of control measures (with constant stringency). As a timeline reference the Oxford Stringency Index [36] is depicted as a colour band from low stringency (darkest blue, index of 0) via intermediate (white, index of 41) to high (darkest red, index of 82).

method provides a way of constructing counterfactual reproduction numbers, which can be useful for monitoring the effectiveness of NPIs across different phases of an epidemic.

Compared to earlier reports on effectiveness of NPIs, we find a lower effectiveness of around 50% during the first wave than reported values around 80% [4,7]. The observed initial basic reproduction number of 2 is lower than other early point estimates ranging from 2.2 to 4.7 [37–41]. Also, the relative transmissibilities of emerging variants that we found are lower than some reported estimates [42,43]. All these differences can be explained by our use of a mean generation interval of 4 days [11,12] instead of the assumed values of 5–7 days in other studies [44–47], as a shorter generation interval leads to lower relative transmissibility of variants, a lower basic reproduction number, and hence a lower effectiveness. This emphasizes the importance of correctly specifying parameters for the natural history of infection within their context and using them consistently in the analysis. For instance, the mean generation interval of 4 days as used in the present study is strongly supported by transmission pair data observed in the early stage of the pandemic in the Netherlands.

The validity of our results can be further assessed by comparing the estimated effectiveness of NPIs against other observations in the Netherlands. One piece of evidence is the agreement in direction between changes in estimated effectiveness and changes in stringency index [36]. Indirect evidence is also provided by the results of contact surveys, as most NPIs aim to reduce contact rates. A contact survey that monitored the contact behaviour in the Dutch general population at regular intervals [48] showed that the periods in which participants had the lowest number of contacts coincided with the periods of the highest effectiveness found in this study. Furthermore, the reconstruction of the immune fractions that was used in the calculations of effectiveness is supported by data. We found good agreement between our results and data on fractions of reinfections and breakthrough infections among test-positive cases, validating the reconstructed immune fractions.

Compared to the first pandemic wave in March/April 2020, more NPIs and stricter NPIs were implemented during the lockdown in January/February 2021. However, for both periods the estimated effectiveness is around 50%, which suggests that the overall impact of the NPIs was lower during the January/February 2021 lockdown. This could be due to differences in behavioural responses across the population, such as over-compliance during the first lockdown (when for example non-essential shops were open but empty) or due to the possibility that other transmission settings (e.g., households) played a larger role. Another explanation could be that heterogeneity in compliance increased during the second lockdown, resulting in more interactions between non-compliers which can increase transmission at the population level.

We assess the impact of three possible sources of bias of the estimated effectiveness. First, the overall immune fraction is a population average of the immune fractions in different age groups. However, if some age groups play a larger role in transmission, for instance due to higher contact rates, such age-specific contributions to transmission should be weighted accordingly [49]. Disregarding this effect indeed overestimated the effectiveness in a simulation study with age stratification but not to a large extent. Second, we assume that all susceptibles are equally susceptible and infectious (once infected), irrespective of whether they have been infected or vaccinated before, while there is evidence to the contrary [50–52]. Differential infectiousness and susceptibility depending on infection and vaccination history could be incorporated in the model, but this would lead to a more complex model structure with additional classes, requiring further data to support such assumption. There is no evidence it would lead to a systematic difference. Third, in reconstructing the immune fraction from the serosurvey data, we did not correct for differences between the vaccination status of survey participants and the general population. The vaccination coverage of the participants was higher than in the general population [20]. Because the vaccine protected them against infection, the cumulative infected fraction of participants could be lower than in the general population. Our reconstruction would yield an underestimation of the immune fraction and hence an overestimation of the effectiveness. This may however bias the results only from July 2021 onwards when a high vaccination coverage was reached in most age groups. Even though these biases can not be excluded, we believe their effect on the results would be limited.

The strength of the proposed method lies in its practical use to track and compare the effectiveness over a long time while avoiding assumptions about a constant effectiveness for each single control measure. Our mechanistic approach allows

for combining various available data sources while propagating the uncertainty in observations, e.g., those in the effective reproduction numbers and the serological survey data. The results show that the uncertainty is sufficiently small for meaningful interpretations of the estimated effectiveness while the outcomes are sufficiently accurate. The method can be applied to any country or region provided that data is available to reconstruct the immune fraction. With appropriate country-specific assumptions (for instance on the seasonal variation in reproduction number), this would allow for a multi-country comparison of effectiveness of measures. The same approach could also be used for other infectious diseases as long as the assumption holds that the counterfactual reproduction number scales linearly with the immune fraction of the population. This would be the case for influenza or other respiratory infections, but not for sexually transmitted infections.

In conclusion, the data-driven approach allowed us to assess the overall effectiveness of NPIs against COVID-19 transmission in the Netherlands over multiple waves of the pandemic. This type of analysis can be used for different stages of an outbreak, epidemic or pandemic. Retrospectively, the results can help to assess the balance between the benefits of the NPIs and the costs, in terms of their impact on, for instance, the economy [53] and general wellbeing [54]. Prospectively, the results can guide public health policy makers in their expectations of the effectiveness of a set of NPIs. With timely estimates on the immune fraction in the population, the approach can be used to monitor effectiveness of NPIs against transmission on a near real-time basis during a pandemic.

## Methods

### Data

The analysis presented here is informed by data collected during the COVID-19 pandemic in the Netherlands. All data is publicly available, either as open data provided by government institutes or as additional data sets on the Github page accompanying this study [26]:

- Population size distribution [55]: the number of persons by age and sex on the first day of every month, from 1 January 2016 until 1 November 2023. The data is aggregated in eight age groups (0–9, 10–19, 20–29, 30–39, 40–49, 50–59, 60–69, 70+) and linearly interpolated between the first day of each month, yielding the population size for each age group at each date.

- Effective reproduction number [11,12]: case reproduction number based on COVID-19 hospitalisations (before 13 June 2020 and after 14 March 2023) or test-positive cases (from 13 June 2020 till 14 March 2023) with 95% confidence interval. The 7-day moving-average is taken to straighten out day-of-week effects.

- Test-positive case data [21,22]: 8.6 million test-positive cases that were reported by the Public Health Services from 28 February 2020 until 31 March 2023. They are aggregated by age group and date of (imputed) symptom onset.

- Genomic surveillance data [19]: weekly number of randomly sampled test-positive cases by SARS-CoV-2 variant from 30 November 2020 onwards; at the end of 2024 the data included almost 180,000 samples.

- Vaccination coverage data [23]: the vaccination coverage of finishing the primary vaccination series (i.e., after one or two vaccine doses depending on the vaccine), by week and 5-year age groups, from 10 January 2021 until 2 January 2022. The vaccination coverage is calculated for our 10-year age groups using the population distribution of 1 January 2022 and linearly interpolated for each date. The data was published online in January 2022 and is now included here as additional data set [26].

- Serological survey data [20,32,33]: the SARS-CoV-2 serological status of a representative sample of the Dutch population in 11 survey rounds from April 2020 until November 2023. Participant numbers range from 3,000–7,000 per survey round. The serostatus of participants is determined either by antibodies against Spike-S1 antigens in unvaccinated persons or by antibodies against Nucleoprotein antigens in vaccinated persons. The cumulative infected fraction of

participants is calculated by age group and survey round with 95% uncertainty. These fractions are included here as additional data set [26], but the raw data is available upon request [56].

- Validation data [22]: For a part of the reported test-positive cases the vaccination status and the infection type (primary infection or reinfection) is known. For weeks in which at least 50% of the cases has a known vaccination status or infection type, the fractions of breakthrough infections and reinfections are calculated to compare with model results. These fractions are included here as additional data sets [26].

- Oxford Stringency Index [36]: index between 0 and 100 indicating stringency of COVID-19 measures by country from 1 January 2020 until 31 December 2022. We use the average index per day in the Netherlands to define periods with constant measures and to compare our results visually.

## Basic reproduction number

The basic reproduction number $R_0(t)$ can vary over time and consists of three parts: the initial basic reproduction number at the start of the pandemic, the seasonal effect and the relative transmissibility of circulating SARS-CoV-2 variants. Before interventions and without any immunity, the observed effective reproduction number is equal to the basic reproduction number $R_0(t)$. Taking the average of the observed effective reproduction number before the reference date $t_{ref}$ of 1 March 2020 we find $R_0^{initial} = 2$.

Seasonal effects can increase $R_0(t)$ in winter due to meteorological conditions that are favorable for virus survival and crowding. The relation between the effective reproduction number and meteorological factors has been studied either by comparing many regions in the world, or by using observations throughout a complete year in one location. The reduction of the reproduction number in summer compared to winter was estimated at 20% in London and Paris [16], 42% in Europe [9,17], and 41% in the Netherlands [18]. Here we assume that seasonal variation $\sigma(t)$ in the reproduction number follows a sinusoid that reaches a maximum at $t_{peak}$ on 1 February [57] at an amplitude of $\alpha = 26\%$, i.e., a 41% reduction in summer compared to winter:

$$\sigma(t) = 1 + \alpha \cos\left(\frac{2\pi}{365}\left(t - t_{peak}\right)\right),$$

where the date difference $(t - t_{peak})$ has a unit of days.

For SARS-CoV-2 the basic reproduction number has also increased over time due to the emergence of new, more transmissible variants. We assume that the increased transmissibility of all variants before the Omicron variant resulted completely in a higher $R_0(t)$, i.e., it was not caused by immune escape. From the genomic surveillance data we estimate the fraction $p_v(t)$ of each variant over time and their relative transmissibility $\theta_v$, defined as the ratio of growth rates of the variant and wildtype (see Section 1 in S1 Text). We calculate the relative transmissibility over time as a weighted average over the variants $\theta(t) = \sum_v \theta_v p_v(t)$.

The time-varying basic reproduction number $R_0(t)$ is the multiplication of the three components, yielding the equation in the main text.

## Reconstruction of immune fraction

To reconstruct the immune fraction of the population over time, we determine the fractions of immunologically naive individuals, immune individuals, and individuals previously infected or vaccinated but not immune, for all age groups 0–9, 10–19, 20–29, 30–39, 40–49, 50–59, 60–69, 70+.

For the infection status, we use a nationwide serological survey that was held every 2–6 months from April 2020 onwards [32]. Because of the longitudinal study design, we can track which participants were infected at least once.

These participants can have been infected multiple times, and they are not necessarily protected against infection. We use this monotonously increasing cumulative infected fraction in each age group as anchor points at the timing of the survey rounds. For our study period we use survey rounds 2, 4, 5 and 6. Rounds 1 and 3 are not used for fitting because the serological test was in development during round 1 and because round 3 followed round 2 too closely. We impute the fraction on the days between survey rounds using the weekly moving average of the reported test-positive case data in the same age group. We assume an incubation period of 5 days [58], and a period of 6 days between symptom onset and detectable serological response [59,60]. We assume an infected person is fully protected against infection for 6 weeks, after which immunity wanes exponentially.

For the vaccination status, we use data that contains the fraction of persons that finished their basic vaccination series (usually after two doses) per week and per age group [23]. This data is linearly interpolated to obtain the vaccinated fraction per day and age group. Up till the moment of finishing the vaccination series a person is assumed to be unprotected. We assume a linear increase from 50% to 100% of the maximal vaccine efficacy in 2 weeks after receiving the last vaccination, which is maintained for 4 weeks before exponential waning starts.

We assume three immunity waning scenarios. For the waning of natural immunity we assume that in the fast/medium/slow waning scenario 49.8%/ 78·6%/ 93.6% of infecteds are still protected 40 weeks after infection [24]. For the waning of vaccine-induced immunity, the slow waning rate is coupled with a high vaccine efficacy and the fast waning rate with a low vaccine efficacy, yielding best-case and worst-case scenarios. We assume that in the fast/medium/slow waning scenario vaccine protection decreases from 80%/ 83%/ 86% maximally to 53%/ 62%/ 69% 18 weeks after finishing the vaccination series [25]. From these data, the exponential waning rates after infection and vaccination are calculated (Table 1, Fig 7 and Section 2.3 in S1 Text).

**Table 1. Overview of parameter values used in the analysis.**

| parameter | value | reference |
|---|---|---|
| initial basic reproduction number | 2 | observed R before 1 March 2020 |
| peak day of seasonally varying reproduction number | 1 February | [57] |
| amplitude of seasonal variation in reproduction number α | 26% | [18] |
| incubation period | 5 days | [58] |
| time between symptom onset and first detectable serogical response | 6 days | [59,60] |
| time between infection and start of waning of immunity | 42 days | assumed equal to start vaccine waning |
| time between finishing vaccination series and maximum vaccine effectiveness | 14 days | [25] |
| time between finishing vaccination series and start of waning of immunity | 42 days | [25] |
| fraction of maximum vaccine effectiveness at finishing vaccination series | 0.5 | assumed |
| maximum vaccine effectiveness | 86% (slow) | [25] |
| | 83% (medium) | |
| | 80% (fast) | |
| waning rate of vaccine-induced immunity $\omega_{vac}$ | 0.002622/day (slow) | derived from [25] |
| | 0.003473/day (medium) | |
| | 0.004902/day (fast) | |
| waning rate of natural immunity $\omega_{inf}$ | 0.000278/day (slow) | derived from [24] |
| | 0.001012/day (medium) | |
| | 0.002929/day (fast) | |

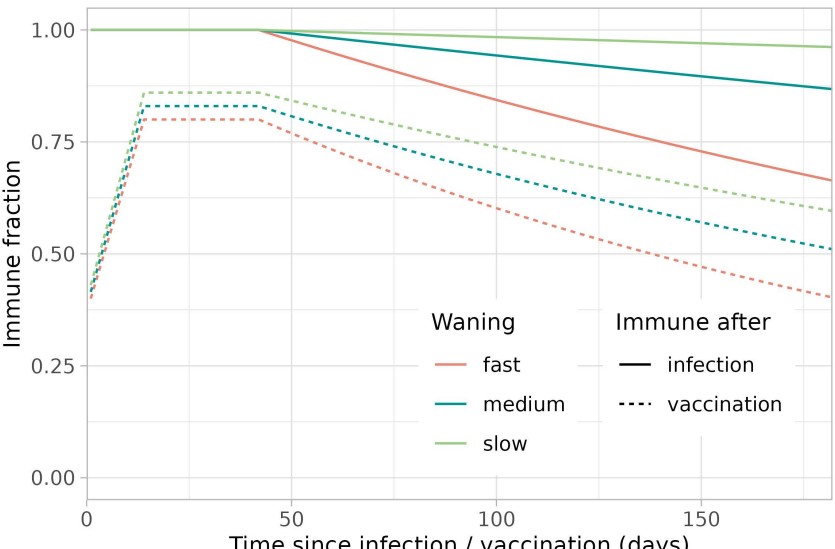

**Fig 7. Waning immunity profiles.** For the waning of immunity after infection and after (finishing the full series of) vaccination, slow, medium and fast waning rates are estimated from literature [24,25].

Each age group is subdivided in classes depending on the infection, vaccination and immune status, where we assume that vaccination is independent of the infection status. For simplicity, we assume all susceptibles within the same age group are equally susceptible, and when infected they are equally infectious. The observed cumulative infected fraction and the assumed waning rates need to be consistent with the true (unobserved) incidence. The reconstruction of the true incidence yields the immune fractions per age group, as explained in detail in Section 2 in S1 Text.

For validation, we calculate the fraction of new infections that were previously infected (reinfections) and vaccinated (breakthrough infections) in the full population. We compare these fractions to the external validation data based on the vaccination status and infection type of reported test-positive cases by plotting them over time. We visually compare the rate and the extent to which the modelled and reported fractions increase.

### Effectiveness calculation

The effectiveness depends on the ratio of the effective and counterfactual reproduction numbers. For both reproduction numbers the 95% confidence interval is known. Assuming they are both normally distributed (i.e., the 95% confidence intervals span 4 standard errors), we approximate the confidence interval of the effectiveness using Fieller's theorem [61].

Table 1 gives an overview of all parameter values that are obtained from data and literature.

### Supporting information

**S1 Text. Supplemental material ''Estimating the effectiveness of non-pharmaceutical interventions against COVID-19 transmission in the Netherlands.''**
(PDF)

### Author contributions

**Conceptualization:** Jantien A. Backer, Don Klinkenberg, Fuminari Miura, Jacco Wallinga.

**Data curation:** Jantien A. Backer.

**Formal analysis:** Jantien A. Backer, Don Klinkenberg, Fuminari Miura, Jacco Wallinga.

**Investigation:** Jantien A. Backer, Don Klinkenberg.

**Methodology:** Jantien A. Backer, Don Klinkenberg, Fuminari Miura, Jacco Wallinga.

**Software:** Jantien A. Backer.

**Supervision:** Jacco Wallinga.

**Validation:** Jantien A. Backer.

**Visualization:** Jantien A. Backer.

**Writing – original draft:** Jantien A. Backer.

**Writing – review & editing:** Jantien A. Backer, Don Klinkenberg, Fuminari Miura, Jacco Wallinga.

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
