## [Decision Letter · Decision Letter 0]

8 Jul 2025

PCOMPBIOL-D-25-00738

Estimating the effectiveness of non-pharmaceutical interventions against COVID-19 transmission in the Netherlands

PLOS Computational Biology

Dear Dr. Backer,

Thank you for submitting your manuscript to PLOS Computational Biology. After careful consideration, we feel that it has merit but does not fully meet PLOS Computational Biology's publication criteria as it currently stands. Therefore, we invite you to submit a revised version of the manuscript that addresses the points raised during the review process.

Please submit your revised manuscript within 30 days Sep 07 2025 11:59PM. If you will need more time than this to complete your revisions, please reply to this message or contact the journal office at ploscompbiol@plos.org. Please include the following items when submitting your revised manuscript:

We look forward to receiving your revised manuscript.

Kind regards,

Philipp Martin Altrock, Ph.D.

Academic Editor

PLOS Computational Biology

Jennifer Flegg

Section Editor

PLOS Computational Biology

**Journal Requirements:**

At this stage, the following Authors/Authors require contributions: Jantien A. Backer. Please ensure that the full contributions of each author are acknowledged in the "Add/Edit/Remove Authors" section of our submission form.

5) We have noticed that you have uploaded Supporting Information files, but you have not included a list of legends. Please add a full list of legends for your Supporting Information files after the references list.

Please state the initials of the authors alongside these funding sources "the Ministry of Health, Welfare and Sport and  European Union’s Horizon research and innovation programme - project ESCAPE".

**Reviewers' comments:**

Reviewer's Responses to Questions

Reviewer #1: Summary

This manuscript presents an approach to estimate the effectiveness of non-pharmaceutical interventions (NPIs) on COVID-19 transmission in the Netherlands from February 2020 to October 2021. By integrating multiple data sources (including reproduction number estimates, genomic surveillance, vaccination coverage, and seroprevalence surveys), the authors simulate time-varying counterfactual reproduction numbers in the absence of NPIs. They quantify the reduction in transmission attributable to NPIs, accounting for three components: (1) seasonality, (2) variant emergence, and (3) population immunity. Overall, the manuscript is methodologically strong and timely. A few clarifications and minor revisions are suggested below.

Major Comments

The counterfactual reproduction number is constructed under the assumption of linear scaling with the immune fraction and multiplicative effects from variant transmissibility and seasonal variation. Please acknowledge that this approach may not fully capture heterogeneities such as contact structure or setting-specific transmission dynamics.

The manuscript introduces a time-varying basic reproduction number, which diverges from the standard epidemiological definition of R0 as a constant. This may cause confusion, especially given the presence of a separate constant “initial R0”. Consider renaming the basic reproduction number to avoid confusion.

Minor Comments

Line 40: Remove “(and references therein)”

Line 110: “lower than winter”

Line 135: “October 2021.”

Figure 1: The 95% confidence intervals for vaccination lines are not available. Either add uncertainty or update the legend to reflect what is shown.

Figure 4: The right panel shows shaded areas for observed breakthrough infections, but the left panel lacks this uncertainty. For consistency, clarify which lines include uncertainty intervals and which do not.

Figures 5 and 6: Display the negative y-axis to reflect effectiveness values below 0%, as mentioned in line 200. Also, include a legend explaining the values represented by the stringency index color band.

Line 280: Suggest explicitly stating the months instead of using “winter 2020/2021”

Line 287: “population level”

Lines 365 and 437: Clarify how validation data were used more clearly.

Table 1: Update the reference column to use the [number] format consistent with in-text citations. Ensure all cited studies are included each row. Specify whether waning rates per day were taken directly from referenced studies or derived by the authors based on those studies.

Reviewer #2: The manuscript titled “Estimating the effectiveness of NPIs against COVID-19 transmission in the Netherlands” quantifies the benefits of NPIs in real time by showing how NPIs lead to decrease in the effective reproduction number.

The manuscript is well-written, and overall I like the work.

I am tempted to suggest acceptance, but would like the authors to respond to some comments/questions.

Minor comments:

Line 62-63: can you further expand on basic reproduction number? By now, there are already three kind of reproduction numbers— effective, counterfactual, and now basic.

Line 109-110: I think you meant “…. In summer 33% lower than in winter”.

Fig. 2: Please briefly describe in the main text how blue curves have been generated? Are these some sort of curve fittings to 3 data points shown in the plots?

In Fig.3: Has any equation being used to plot the curves? If so, please mention it in the main text. Or is it purely data analysis and interpolation?

Line 142-144: Please briefly add to the text how you have accounted for immunity waning. For a PloS CB reader, I would tend to expect a bit more detail description in the main text.

Line 145: may be consider changing naïve to naive (for consistency)

Line 152: Please cite some references to support this. Or is it the conclusion from Fig. 3 that vaccine induced immunity wanes faster than natural immunity?

Line 163: What are these “calculations”? Are these mathematical calculation? Please clarify on this.

Line 164: What are breakthrough infections and how are they different from reinfections?

Line 169: The sentence suggests that there is some underlying model for immunity waning. Can you bring it to the main text and discuss here?

Line 193-194: Is it a new result? If not, it will be helpful to cite some references. Same for the sentence 197-198.

Line 199: What does negative effectiveness mean? How can NPIs have negative effect?

Line 216: Are you particularly referring to Fig. 5B? If so, please update the figure reference accordingly.

General comments:

Why only restrict to Netherlands? How about comparison with the results of refs. 4-9 that assume constant affect of season etc. through time?

The manuscript is about quantifying (important) the effectiveness of NPIs. We all in some sense agree about the usefulness of NPIs. But what is the cost of NPIs? Can you say anything about that? Example, economic cost, social resistance etc. Any references?

Line 106: For other variants immune escape is also at play. Yet R_0 is defined for those strains. Is it true? Shouldn’t immune escape be somehow taken care by the definition of R_0?

**Have the authors made all data and (if applicable) computational code underlying the findings in their manuscript fully available?**

Reviewer #1: Yes

Reviewer #2: Yes

PLOS authors have the option to publish the peer review history of their article (what does this mean? ). If published, this will include your full peer review and any attached files.

**Do you want your identity to be public for this peer review?** For information about this choice, including consent withdrawal, please see our Privacy Policy .

Reviewer #1: No

Reviewer #2: No

**Figure resubmission:**
---

## [Decision Letter · Decision Letter 1]

8 Sep 2025

Dear Dr. ir. Backer,

We are pleased to inform you that your manuscript 'Estimating the effectiveness of non-pharmaceutical interventions against COVID-19 transmission in the Netherlands' has been provisionally accepted for publication in PLOS Computational Biology.

Best regards,

Philipp Martin Altrock, Ph.D.

Academic Editor

PLOS Computational Biology

Jennifer Flegg

Section Editor

PLOS Computational Biology

Reviewer #1:

Reviewer's Responses to Questions

**Comments to the Authors:**

Reviewer #1: All comments have been addressed.

**Have the authors made all data and (if applicable) computational code underlying the findings in their manuscript fully available?**

Reviewer #1: Yes

PLOS authors have the option to publish the peer review history of their article (what does this mean? ). If published, this will include your full peer review and any attached files.

**Do you want your identity to be public for this peer review?** For information about this choice, including consent withdrawal, please see our Privacy Policy .

Reviewer #1: No

---

## [Editor Report · Acceptance letter]

PCOMPBIOL-D-25-00738R1

Estimating the effectiveness of non-pharmaceutical interventions against COVID-19 transmission in the Netherlands

Dear Dr Backer,

I am pleased to inform you that your manuscript has been formally accepted for publication in PLOS Computational Biology. Your manuscript is now with our production department and you will be notified of the publication date in due course.

With kind regards,

Zsofia Freund
